# Enhancement of the Synaptic Performance of Phosphorus-Enriched, Electric Double-Layer, Thin-Film Transistors

Dong-Gyun Mah [1], Hamin Park [2] and Won-Ju Cho [1,*]

1   Department of Electronic Materials Engineering, Kwangwoon University, Gwangun-ro 20, Nowon-gu, Seoul 01897, Republic of Korea; madong13@kw.ac.kr
2   Department of Electronic Engineering, Kwangwoon University, Gwangun-ro 20, Nowon-gu, Seoul 01897, Republic of Korea
*   Correspondence: chowj@kw.ac.kr; Tel.: +82-2-940-5163

**Abstract:** The primary objective of neuromorphic electronic devices is the implementation of neural networks that replicate the memory and learning functions of biological synapses. To exploit the advantages of electrolyte gate synaptic transistors operating like biological synapses, we engineered electric double-layer transistors (EDLTs) using phosphorus-doped silicate glass (PSG). To investigate the effects of phosphorus on the EDL and synaptic behavior, undoped silicate spin-on-glass-based transistors were fabricated as a control group. Initially, we measured the frequency-dependent capacitance and double-sweep transfer curves for the metal-oxide-semiconductor (MOS) capacitors and MOS field-effect transistors. Subsequently, we analyzed the excitatory post-synaptic currents (EPSCs), including pre-synaptic single spikes, double spikes, and frequency variations. The capacitance and hysteresis window characteristics of the PSG for synaptic operations were verified. To assess the specific synaptic operational characteristics of PSG-EDLTs, we examined EPSCs based on the spike number and established synaptic weights in potentiation and depression (P/D) in relation to pre-synaptic variables. Normalizing the P/D results, we extracted the parameter values for the nonlinearity factor, asymmetric ratio, and dynamic range based on the pre-synaptic variables, revealing the trade-off relationships among them. Finally, based on artificial neural network simulations, we verified the high-recognition rate of PSG-EDLTs for handwritten digits. These results suggest that phosphorus-based EDLTs are beneficial for implementing high-performance artificial synaptic hardware.

**Keywords:** electric double-layer transistors; phosphorus-doped silicate glass; artificial neural network; excitatory post-synaptic currents; recognition rate





## 1. Introduction

Advancements in semiconductor technology have found applications in diverse industries and have thus enriched human life. Notably, these advancements have facilitated the development of artificial intelligence (AI) technologies capable of learning, reasoning, and problem-solving attributes that closely emulate the capabilities of human-like computer systems [1–5]. However, the conventional von Neumann architecture, initially proposed for computing devices, presents challenges in AI implementation owing to issues such as memory bottlenecks, speed limitations, and energy inefficiency [6–9]. Neuromorphic computing systems, which integrate memory and computational functions by connecting chips in parallel, have been explored as viable alternatives [10–12]. The key lies in optimizing hardware implementations to mimic the operational mechanisms of biological neurons and synapses [13,14]. Various devices, including charging-based field-effect, ferroelectric field-effect, electric double-layer, and electrochemical transistors, have been identified for hardware implementation [15–20].

In particular, we focused on the electrolyte-based EDL characteristics used as the gate dielectric layer in TFTs. Firstly, among the representative classifications of TFTs utilizing

the EDL characteristics of the electrolyte membrane are polyelectrolytes, ionic liquids, ionic gels, and solid electrolytes. Particularly, solid electrolytes have drawn our attention due to their relatively stable chemical properties and lower production costs. To elaborate further, solid electrolytes are classified into organic, inorganic, and organic–inorganic hybrids, finding applications in synaptic transistors and future prospects. Here, we specifically focused on the chemical stability of inorganic solid electrolytes. Notably, phosphosilicate glass demonstrates the advantage of a strong lateral EDL capacitor due to internal proton movement and has been extensively researched as a material for synaptic applications. While conventional methods often employ PECVD processes for PSG-based EDLT fabrication, we observed the advantages of PSG produced using solution-based spin-coating techniques, including low-process costs, time efficiency, and effective control over the internal phosphorus concentration. Through a successful comparison with devices using pure silicate glass without phosphorus doping, we have successfully confirmed the potential of electric double-layer transistors (EDLTs) using phosphorus-doped silicate glass (PSG) [21–25].

To assess the effects of phosphorus on the EDL and synaptic behavior, undoped silicate spin-on glass (SOG)-based transistors were prepared for comparative analyses [26,27]. The electrical properties of the two types of silicate glass layers were analyzed by measuring the frequency-dependent (C–*f*) capacitance curve using a metal-oxide-semiconductor (MOS) capacitor configuration [28,29]. PSG-based MOS capacitors exhibited high-capacitance values ranging from 18.666 $\mu$F/cm$^2$ to 0.109 $\mu$F/cm$^2$ in the frequency range of 1–10$^7$ Hz. In contrast, SOG-based MOS capacitors have low capacitances in the range of 0.023–0.016 $\mu$F/cm$^2$ in the same frequency range. The double-sweep transfer curve of the MOS field-effect transistor was measured by increasing the maximum gate voltage to determine the tendency of the hysteresis window and maximum drain current induced by P-doping [30,31]. In addition, to verify the synaptic properties of the proposed PSG-based EDLT device, we measured the excitatory post-synaptic current (EPSC) with a single spike, double spike, and at different frequencies [32,33]. The results showed that the PSG exhibited excellent synaptic properties, such as synaptic weighting, learning, and memory emulation, depending on the spiking conditions; by contrast, the SOG hardly exhibited these synaptic properties. We also examined the changes in synaptic weight during potentiation and depression as a function of the number of presynaptic spikes. Systematic potentiation and depression (P/D) measurements were conducted by varying their amplitude and duration. We found that the normalized P/D results facilitated parametric extraction of the nonlinearity factors, asymmetry ratio (AR), and dynamic range (DR) based on presynaptic variables, thus revealing trade-off relationships [34]. Finally, using the modified National Institute of Standards and Technology (MNIST) dataset, we obtained the recognition rate of the PSG-based EDLTs for handwritten numbers using artificial neural network (ANN) simulations [35]. Achieving a recognition rate (>90%) across various pulse conditions, especially at the optimal spike count (N = 30), demonstrated the potential suitability of PSG-based EDLTs for artificial synapse hardware network configurations in terms of low power consumption. In conclusion, phosphorus-based electrolytes offer the potential for the implementation of artificial synapses, and the proposed PSG-EDLT is promising for the development of efficient artificial synaptic systems.

## 2. Materials and Methods

### 2.1. Material Specifications

The materials employed to fabricate the phosphorus-doped silicate glass (PSG) and undoped silicate glass included 20B spin-on glass (SOG) (Filmtronics Inc., Butler, PA, USA) and P509 spin-on dopant (SOD) (Filmtronics Inc., Butler, PA, USA). Specifically, for undoped silicate glass, we utilized 20B SOG, whereas the PSG fabrication involved 20B SOG and P509 SOD. Transistors and MOS capacitors for each material were fabricated using an indium-gallium-zinc-oxide (IGZO) sputter target (In$_2$O$_3$:Ga$_2$O$_3$:ZnO = 4:2:4.1 mol%, THIFINE Co., Ltd., Incheon, Republic of Korea), indium-tin-oxide (ITO) film (In$_2$O$_3$:SnO$_2$ = 9:1 mol%, THIFINE Co., Ltd., Incheon, Republic of Korea), and aluminum (Al) pellet (purity > 99.999%;

THIFINE Corp., Incheon, Republic of Korea). Additionally, a p-type Si substrate (plane, (100); resistivity range of 1–10 $\Omega \cdot$cm; LG Siltron Inc., Gumi, Republic of Korea) was employed in the fabrication process.

### 2.2. Synthesis of PSG and SOG Films

To prepare a PSG electrolyte film doped with phosphorus (10.5% phosphorus concentration compared with the total mass of the formed PSG film), we employed a blend of SOG and SOD solutions. In this experiment, we fabricated a PSG electrolyte film with notable electric double-layer (EDL) characteristics with an SOG and SOD blending mass ratio of 3:7. Pure SOG was used for the silicate film fabrication, and the same heat treatment process as that used for PSG was applied. The SOG and SOD solutions consisted of ethanol ($C_2H_6O$) and water ($H_2O$). Additionally, the SOG included 2-propanol ($C_3H_8O$), acetone ($CH_3COCH_3$), and polysilicate polymers. Depending on the heat-treatment conditions, such as the temperature, time, and ambient gas, impurities and solvents were removed from the interior of the SOG. SOG treated at high temperatures for an adequate duration exhibited excellent insulation properties akin to $SiO_2$, thereby enhancing its electrical stability [36]. SOD comprised components of a phosphorus silicate polymer with a phosphorus mass ratio of 15% of the total mass ratio of SOD. By using a high-temperature process, impurities were expelled from the internal parts, forming an insulating layer similar to that of the P-doped $SiO_2$ by creating P-O and P-OH groups [37]. As all the solutions tend to solidify at room temperature, they were stored at temperatures lower than $-2$ °C. We employed spin coating to blend SOD and SOG; additionally, magnetic stirring at room temperature for 8 h was used to enhance the uniformity of the internal phosphorus mixture and prevent issues related to thick films and condensation [38].

### 2.3. MOS Capacitor and Transistor Fabrication Using PSG and SOG Films

To assess the EDL characteristics of the PSG film, we fabricated PSG-MOS capacitors, SOG-MOS capacitors, PSG transistors, and SOG transistors to compare the electrical synaptic properties of the PSG and SOG. Initially, the P-type Si substrates used as gates were cleaned in all cases using the wet-chemistry-based standard Radio Corporation of America (RCA) cleaning process. In the PSG case, the blended SOG and SOD solutions were spin-coated onto a $1 \times 1$ cm$^2$ p-Si substrate at 6000 revolutions per minute for 30 s. Similar spin-coating conditions were applied for SOG solutions. Subsequently, pre-baking was conducted on a hot plate with the following temperature profile: 70 °C for 3 min, 100 °C/150 °C/200 °C for 2 min each. The blended solution film was then cured, and impurities were removed by thermal annealing performed at 650 °C for 1 h in a forming gas (5% $H_2$ + 95% $N_2$); this resulted in ~300 nm-thick PSG and ~250 nm-thick SOG films. For the MOS capacitors, a 150 nm-thick Al film was deposited using an e-beam. In the case of the transistors, a 50 nm-thick IGZO channel layer with dimensions (width $\times$ length) of 120 μm $\times$ 60 μm was deposited using radio frequency (RF) magnetron sputtering. Finally, a 150 nm-thick ITO film was deposited using RF sputtering and lifted off to form source/drain (S/D) electrodes with dimensions (width $\times$ length) of 150 μm $\times$ 120 μm.

### 2.4. Characterization Method

The fabricated devices were stored in a humidity-controlled environment to prevent impurity absorption or chemical reactions caused by external humidity, which could alter the EDL characteristics. In addition, to measure the capacitances of the Al/PSG and SOG/p-Si MOS capacitors at various frequencies, we used an Agilent 4284A precision LCR meter (Hewlett–Packard Corporation, Palo Alto, CA, USA). The EDLT curves were characterized using an Agilent 4156B precision semiconductor parameter analyzer (Hewlett-Packard Co., Palo Alto, CA, USA). Furthermore, two identical Agilent 8110A pulse generators (Hewlett-Packard Co., Palo Alto, CA, USA) were jointly used to apply electrical pre-synaptic and post-synaptic stimuli to verify the synaptic operation of the EDLT.

## 3. Results and Discussion

### 3.1. Electrical Characteristics of PSG and SOG-Based MOS Capacitors

To evaluate the synaptic properties of the PSG and SOG electrolyte films, MOS capacitors with Al/PSG/p-Si and Al/SOG/p-Si structures with Al electrode diameters of 200 μm were fabricated [39]. Synaptic devices typically mimic biological synaptic features by applying stimuli in the low-frequency range of 1–100 Hz to observe synaptic behavior. Figure 1a presents the frequency-dependent (C–$f$) capacitance curves measured in the range of 1–$10^7$ Hz. In the case of the PSG-MOS capacitor, the capacitance showed a gradual increase at higher frequencies; however, at frequencies lower than $10^3$ Hz, there was a notable increase, especially at 1 Hz, where it reached the value of 18.64 μF/cm². The significant increase in capacitance at low frequencies suggests excellent EDL characteristics within the PSG for implementing synaptic features [40]. However, the SOG-MOS capacitor maintained a maximum value of 0.02 μF/cm² regardless of frequency, thus showing a distinct difference compared with that obtained in the PSG case. Figure 1b illustrates the schematic and measurement methods for PSG- and SOG-based transistors, as well as the operational mechanisms mimicking the behavior of biologically inspired neuromorphic systems. Using the approach outlined in Figure 1b, we conducted a detailed analysis of proton movements between the insulating layer and channel under various electrical stimulation conditions [41,42]. This allowed the specific evaluation of the EDL characteristics and synaptic behaviors of the two types of components. These findings underscore the unique synaptic properties of PSG and offer promising implications for the development of neuromorphic systems with enhanced functionality and efficiency. The observed capacitance variations in the PSG-MOS capacitors, especially the substantial increase at low frequencies, indicate a pronounced sensitivity to electrical stimuli. This heightened sensitivity aligns with the desired characteristics for emulating biological synapses. In contrast, the SOG-MOS capacitor exhibited consistent capacitance values regardless of frequency, thus suggesting limited sensitivity to electrical stimuli. This disparity in behavior underscores the distinct synaptic responses of the PSG and SOG components. Furthermore, the specific differences in the internal characteristics of the PSG and SOG films under the influence of an external electric field can be observed through Figure 1b. For SOG, it is based on the formation of a Si-O network, representing the characteristics of silicon dioxide. However, in the case of PSG, when an electric field is applied, O-H bonding readily dissociates from the P-OH groups, leading to the formation of mobile protons for EDL formation. These protons accumulate at the interface between the channel and PSG through continuous hopping.

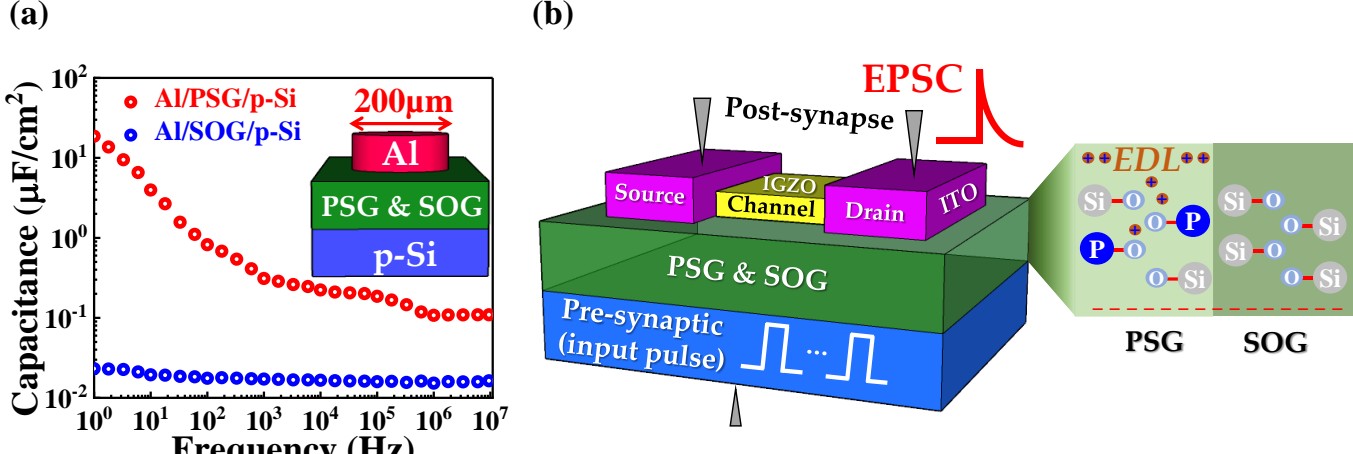

**Figure 1.** (a) Capacitance–frequency (C–$f$) curves of aluminum (Al)/phosphorus-doped silicate glass (PSG) and silicate spin-on glass (SOG)/p-Si metal-oxide semiconductor (MOS) capacitors. The inset in (a) illustrates the MOS capacitor structure. (b) Schematic of the proposed PSG and SOG-based transistor, elucidating the mechanism of the PSG-EDL.

### 3.2. Comparison of EDL Operation between PSG and SOG-Based Transistors

Evaluating the electrical characteristics of the EDLT is crucial as it forms the basis for mimicking the biological mechanisms of the brain [43]. Therefore, we measured and analyzed the transfer characteristics ($I_D$–$V_G$) of both transistor types. Figure 2a shows the $I_D$–$V_G$ curves of the PSG-based transistor at the maximum bottom-gate voltage (max. $V_G$) increases from 1 V to 6 V at 0.5 V intervals. We observed a counterclockwise hysteresis window and found consistency in the transistor's turn-on threshold voltage with changes in the max. $V_G$, thus ensuring the reliability of the operation [44]. The inset in Figure 2a represents the maximum drain current (max. $I_D$) in the IGZO channel with increasing $V_G$. As the max. $V_G$ increases, the max. $I_D$ increases from 8.56 μA to 25.71 μA. This was attributed to the enhanced EDL characteristics at the PSG layer and channel interface following incremental increases in the max. $V_G$, thus yielding a linear increase ($R^2$ = 95.09). Figure 2b illustrates the $I_D$–$V_G$ curves of the SOG-based transistors as the max. $V_G$ increases from 1 to 6 V. It is difficult to identify the counterclockwise hysteresis as the max. $V_G$ increases, and the turn-on threshold voltage of the transistor becomes unstable. Furthermore, comparing the insets of Figure 2a,b, the slope of the max. $I_D$ increases as a function of the max. $V_G$; additionally, a significant difference (an approximate 8.25 times difference) is observed between PSG (3.3 μA/V) and SOG (0.4 μA/V). Furthermore, from the TFT perspective, when calculating the mobility based on the forward direction of the $V_G$ sweep (−6 to 6 V), PSG showed a value of 28.54 $cm^2V^{-1}s^{-1}$, while SOG exhibited a value of 19.57 $cm^2V^{-1}s^{-1}$. This confirms a relatively higher mobility for PSG, attributed to the EDL effect between the channel and the interface. Figure 2c represents the hysteresis window voltage with respect to the max. $V_G$. In the case of PSG, the rate of change in the hysteresis window (from 1.19 V to 4.6 V) yielded a slope of 0.75 *v/v*, while for SOG, the hysteresis window change (almost incapable of confirming EDL characteristics (from 0.13 V to 0.44 V)) yielded a slope of 0.07 *v/v*. In Figure 2d, the $I_D$-$V_D$ output curves measured at regular intervals as |$V_G$-$V_{th}$| increases from 0 to 6 V are depicted. PSG exhibits a greater drain current value compared to SOG because it has an EDL gate effect within the electrolyte layer. Additionally, $I_D$ increases linearly with the increase in $V_D$, and saturation is observed as $V_D$ increases, indicating typical pinch-off characteristics. These detailed analyses provide insights into the distinct EDL operation characteristics of PSG- and SOG-based transistors, emphasizing the significance of PSG in achieving reliable and consistent synaptic behavior for neuromorphic applications.

Furthermore, achieving synaptic characteristics in EDLTs requires an excellent SS and current on/off ratio in the transistor itself. A better SS facilitates the rapid movement of internal charges in response to pulse application, and a superior current on/off ratio reduces the power consumption for pulse application. Therefore, we conducted a comparison of characteristics for solid electrolyte-based EDL transistors (Table 1), and the fabricated devices exhibited relatively superior properties with an SS value of 106 (mV/dec) and current on/off ratio of $3.5 \times 10^6$ [45–48].

**Table 1.** Performance of EDL transistors using the IGZO channel based on solid-state electrolytes.

| Channel Layer | Electrolyte | SS (mV/dec) | On/Off Ratio | Year | Ref. |
|---|---|---|---|---|---|
| IGZO | Mesoporous $SiO_2$ | 110 | $1.1 \times 10^6$ | 2009 | [45] |
| IGZO | Porous $SiO_2$ | 130 | $10^5$ | 2017 | [46] |
| IGZO | Chitosan | 162 | $2.7 \times 10^6$ | 2021 | [47] |
| IGZO | PSSNa | 157 | $1.5 \times 10^4$ | 2024 | [48] |
| IGZO | P-doped silicate glass | 106 | $3.5 \times 10^6$ | 2024 | This work |

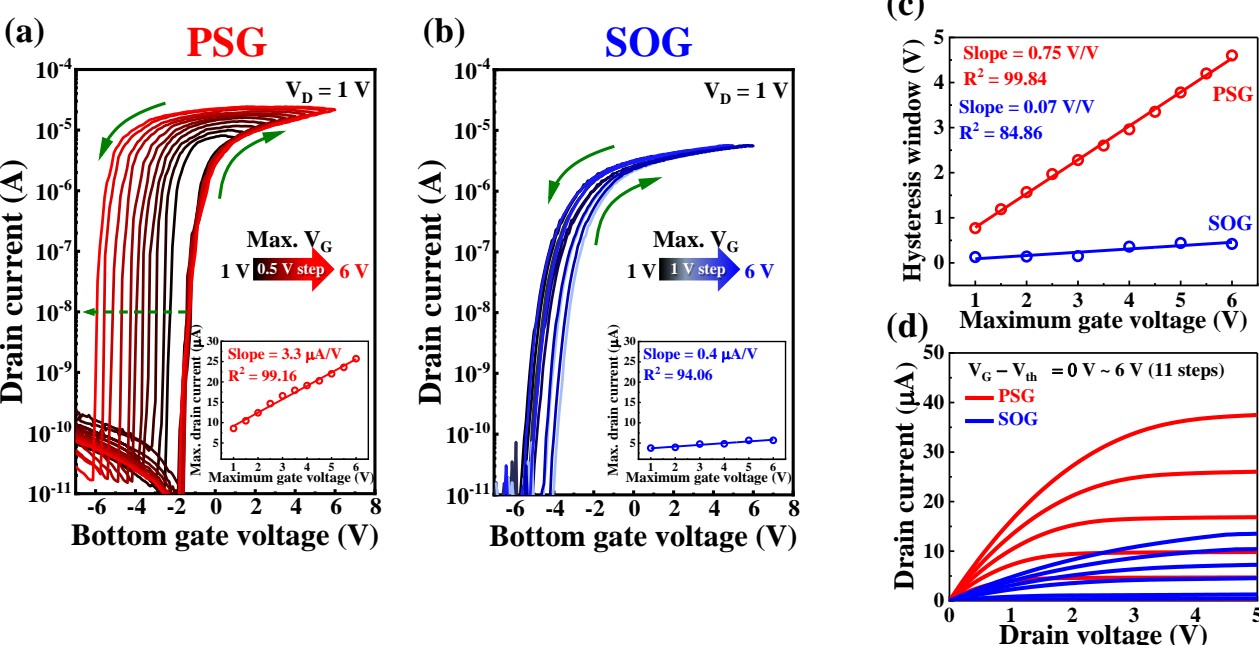

**Figure 2.** (**a**) Double-sweep transfer curves versus maximum gate voltage (1 V to 6 V in steps of 0.5 V) at a constant $V_D = 1$ V in the PSG transistor case. (**b**) Double-sweep transfer curves versus maximum gate voltage (1 V to 6 V in steps of 1 V) at a constant $V_D = 1$ V in the SOG transistor case. The insets in (**a**,**b**) denote the maximum drain current as increasing maximum $V_G$ values. (**c**) Hysteresis window voltage plots as a function of the maximum $V_G$ for the PSG and SOG. (**d**) Output curves ($I_D$–$V_D$) of the SOG and PSG transistors.

### 3.3. Comparison of Synaptic Characteristics between PSG and SOG Transistors

The biological operational mechanisms of the brain are based on the intricate relationship between neurons and synapses [49]. Synapses play a crucial role as connection sites, transmitting information between neurons, and their chemical signal strengths change owing to various factors. As depicted in Figure 1b, the input pulses serve the pre-synaptic role of altering the strength of synapses, and the EPSC serves as an indicator representing the post-synaptic result in the form of electrical signals. Initially, for both types of device, the duration ($\Delta t$) of a single spike was increased from 10 ms to 900 ms using an amplitude of 1 V and $V_D = 1$ V. Figure 3a illustrates the EPSC results for PSG-based transistors with a single spike. The maximum EPSC value significantly increases as a function of the duration, and the characteristics of the post-synaptic residual region are also enhanced. Figure 3b presents the EPSC results for the SOG-based transistors with a single spike. In contrast to PSG, the synaptic characteristics of memory and learning in the post-synaptic residual region are not evident, and EPSC saturation occurs from 500 ms onward. Figure 3c shows the results for the maximum EPSC values from 10 ms to 900 ms for each studied case, confirming the excellence of the phosphorus-based EDLT device functionality under single-spike conditions. These findings indicate the superior synaptic performance of PSG-based transistors compared with that of SOG-based transistors and emphasize the potential of phosphorus-based EDLTs for realizing efficient artificial synaptic systems.

One of the specific phenomena occurring in synapses is the facilitation of the second stimulus (induced by the first stimulus) when two pre-synaptic spikes are applied consecutively [50,51]. To observe the differences in paired-pulse facilitated phenomena between PSG and SOG, measurements were conducted with consecutive pulses at $\Delta t_{interval}$ of 50 ms, 400 ms, and 1.9 s. Figure 4a,d represent the EPSC results of PSG and SOG, respectively, with two consecutive pulses at a short time interval of 50 ms. Comparing the EPSC value induced by the second pulse to that induced by the first pulse ($A_2/A_1$), PSG and SOG yielded superior responses of 270.1% and 105.9%, respectively, thus indicating the excellent

weighted effect of PSG. Figure 4b,e depict the results for a time interval of 400 ms. Specifically, for PSG, the synaptic weight effect based on consecutive pulses appeared weaker compared to the 50 ms case, and a clearer distinction between the boundaries of the first and second spikes was observed. However, PSG exhibited a relatively higher value of 167.7% compared to the $A_2/A_1$ value of 103.6% for SOG. Figure 4c,f show the EPSC results for the PSG and SOG, respectively, with two consecutive pulses at a relatively long time interval of 1.9 s. The weighted effect due to consecutive stimuli decreased as the time interval increased, with PSG and SOG values of 103.8% and 100.6%, respectively. In conclusion, the sensitive response reactivity of phosphorus-based electrolyte films can be artificially enhanced by using appropriate consecutive pulse application intervals. Characteristic analysis was also conducted with changes in frequency and an increase in the number of consecutive pulse applications [52,53].

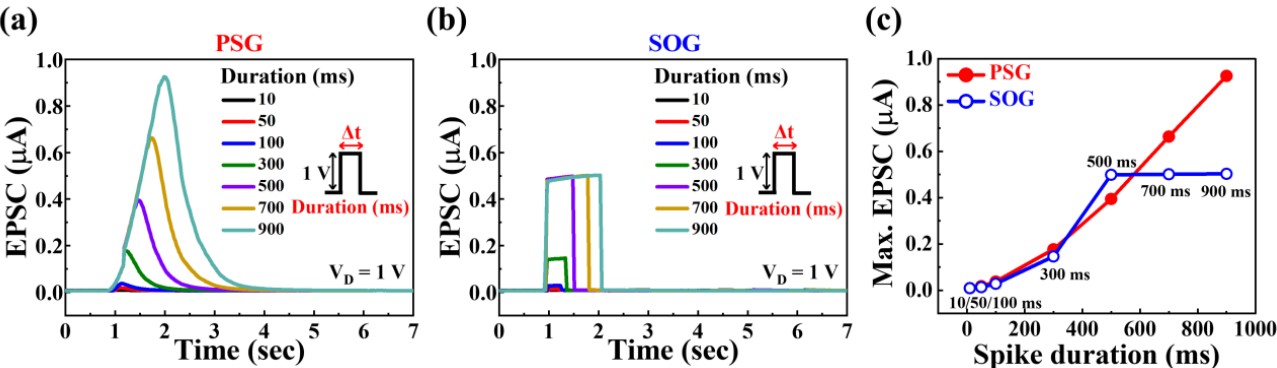

**Figure 3.** Excitatory post-synaptic currents (EPSCs) are generated by a single spike with a fixed amplitude (1 V) at various durations (10 to 900 ms) for (**a**) PSG transistor and (**b**) SOG transistor. (**c**) Maximum EPSC values from 10 ms to 900 ms.

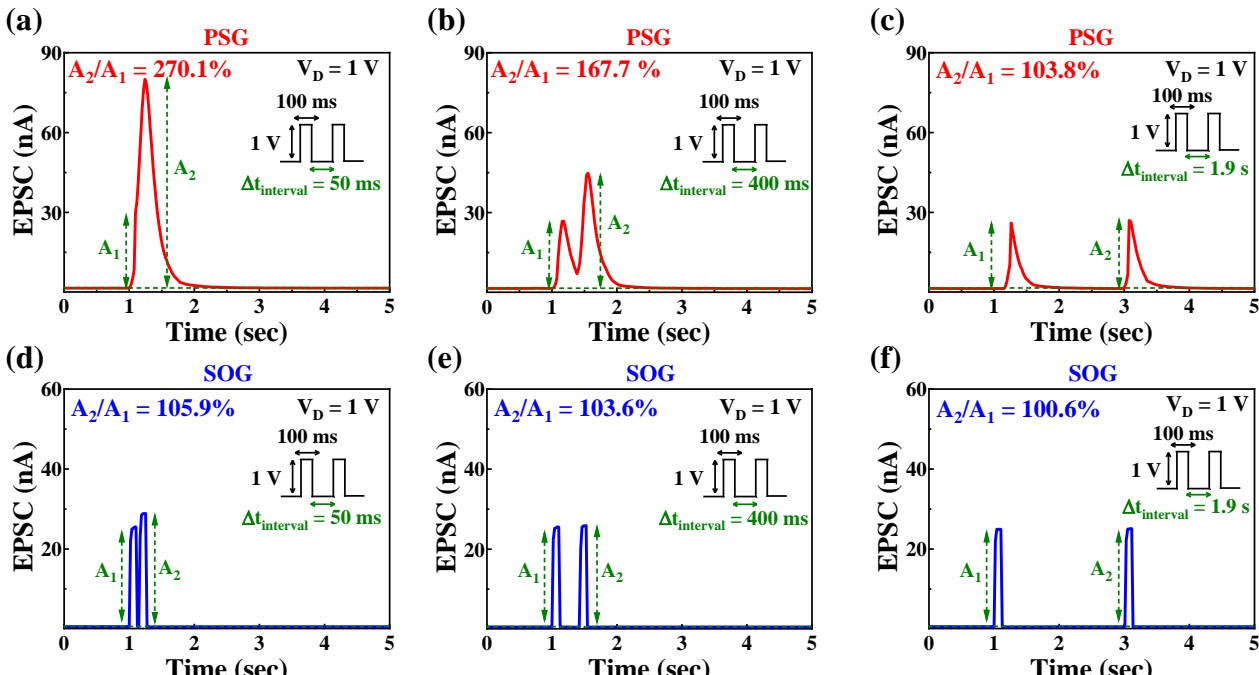

**Figure 4.** Paired-pulse facilitated EPSC by a paired pre-synaptic spike (1 V, 100 ms) at $\Delta t_{interval}$ = 50 ms for (**a**) PSG and (**d**) SOG, at $\Delta t_{interval}$ = 400 ms for (**b**) PSG and (**e**) SOG and at $\Delta t_{interval}$ = 1.9 s for (**c**) PSG and (**f**) SOG.

Finally, for both types of devices, EPSCs were measured in response to pre-synaptic stimulations in the low-frequency range of 1–9.8 Hz using a fixed spike number (N = 10), amplitude (1 V), and duration (100 ms). Figure 5a illustrates the variation in the EPSC with respect to the PSG frequency. As the frequency increases from 1 to 9.8 Hz, the maximum EPSC value of the 10th spike increases by approximately 37.7-fold (from 0.03 µA to 1.13 µA). In contrast, Figure 5b shows that for the SOG, the synaptic weight effect induced owing to the variation in frequency was minimal, and the characteristics of the residual region were also unfavorable. For a quantitative analysis, the synaptic weights were compared, as shown in Figure 5c. The calculation involved the subtraction of the initial EPSC ($G_0$) from the maximum value before the application of the spike stimulus. The EPSC ($G_{10}$) value for the 10th spike was divided by the time interval from $G_1$ to $G_{10}$. As a result, PSG showed a tendency to increase from 0.004 (µA/sec) to 1.087 (µA/sec) ($R^2$ = 98.34), while SOG reached a maximum of 0.029 (µA/sec), thus indicating a lower synaptic weight. In conclusion, the advantages of PSG-EDLT were highlighted by comparing the SOG and PSG values. Additional measurements were performed to explore various synaptic characteristics using PSG-EDLT and implement artificial synapses.

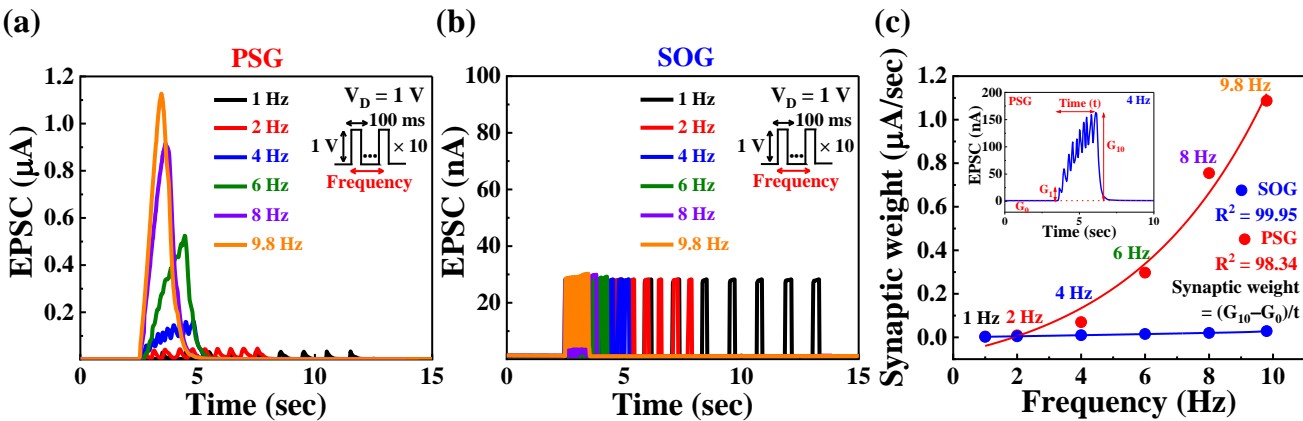

**Figure 5.** EPSC responses generated by consecutive pre-synaptic spikes (1 V, 100 ms) at various frequencies from 1 to 9.8 Hz for (**a**) PSG and (**b**) SOG. (**c**) Pre-synaptic spike-frequency-dependent synaptic weight. Inset denotes the EPSC response to 4 Hz in the PSG device case.

### 3.4. Superiority of Phosphorus-Based PSG as an EDLT

Evaluating the characteristics of long-term potentiation (LTP) and depression (in PSG-EDLT) is crucial for the practical implementation of artificial synaptic devices in neural networks. Therefore, we initially measured the EPSC by increasing the pulse number from 1# to 40#, with the pre-synaptic conditions of an amplitude of 1 V, duration of 100 ms, and an inter-pulse interval of 10 ms. In Figure 6a, the results of the EPSC in response to the pre-synaptic stimuli show an approximate 38.2-fold increase from 0.06 µA to 2.29 µA as the pulse number increases. Figure 6b shows the characteristics of the PSG-EDLT with increasing spike numbers, thus displaying a high correlation ($R^2$ = 99.95), indicating a gradual saturation of the maximum EPSC value. Subsequently, we conducted additional analyses of the P/D characteristics for spike numbers ranging from 10# to 50# (in 10# increments) by focusing on variations in channel conductance [54,55].

Figure 7a presents the P/D results obtained by varying the pre-synaptic pulse numbers (10#/20#/30#/40#/50#) and by applying a read pulse ($V_D$ = 1 V, duration = 100 ms) and an input pulse ($V_G$ = 1 V, duration = 100 ms). With an increasing number of pulses, distinctive characteristics in terms of maximum conductance and linearity became apparent. Nonlinearity serves as a critical indicator of the relationship between the input and output in LTP and LTD and plays a pivotal role in simulating neural network learning and pattern recognition. To analyze the relationship between the maximum conductance and

nonlinearity in the P/D results based on the pulse numbers, we quantified the nonlinearity factor using the following equation,

$$G = \begin{cases} \{(G_{max}{}^{\alpha} - G_{min}{}^{\alpha}) \times w + G_{min}{}^{\alpha}\}^{1/\alpha} & if\ \alpha \neq 0, \\ G_{min} \times (G_{max}/G_{min})^{w} & if\ \alpha = 0. \end{cases} \quad (1)$$

where $G_{max}$ and $G_{min}$ represent the maximum and minimum conductance, respectively, and w is an internal variable ranging from 0 to 1 [56]. The nonlinearity coefficient $\alpha$ governs either potentiation ($\alpha_p$) or depression ($\alpha_d$), with the ideal value for the nonlinearity factor being 1. As shown in Figure 7b,c, the P/D nonlinearity factors increased as a function of the pulse numbers and reached values in the ranges of 1.65 to 2.7 and $-3.5$ to $-0.35$, respectively, deviating from the ideal value of 1. Conversely, the maximum conductance increased from 0.51 µS to 0.99 µS. This observation indicates a trade-off relationship between the nonlinearity factors and conductance during potentiation and depression. Subsequently, an extension of these findings reveals that under the conditions of 30#, the ANN simulation achieves the highest recognition rate. Therefore, additional P/D measurements were conducted with the amplitude and duration as variables under the conditions of 30#.

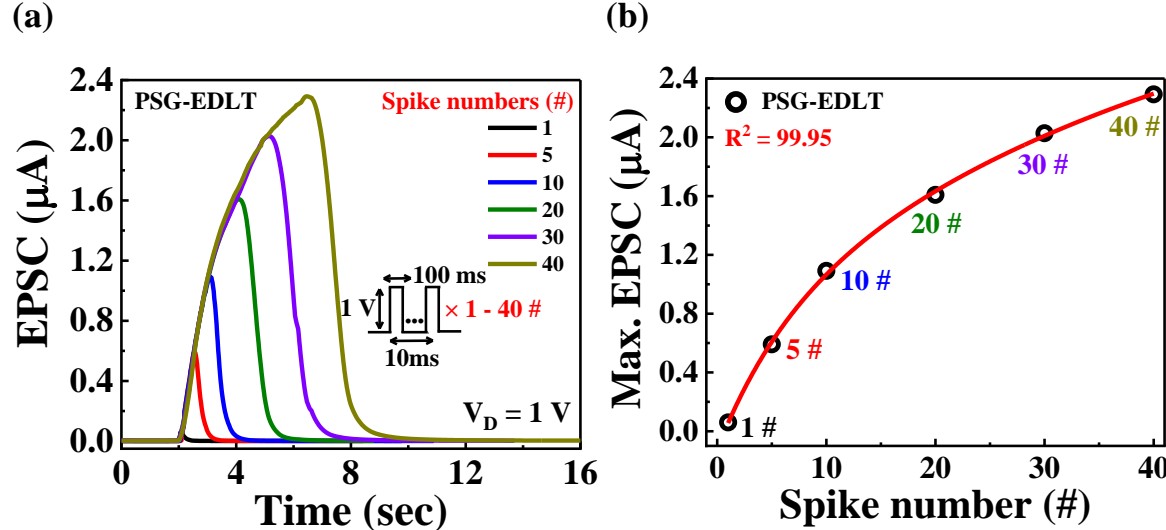

**Figure 6.** (**a**) EPSC responses generated by varying the number of pre-synaptic spikes (1 V, 100 ms) for PSG-EDLT. (**b**) Maximum EPSC values from the 1st to the 40th pulse.

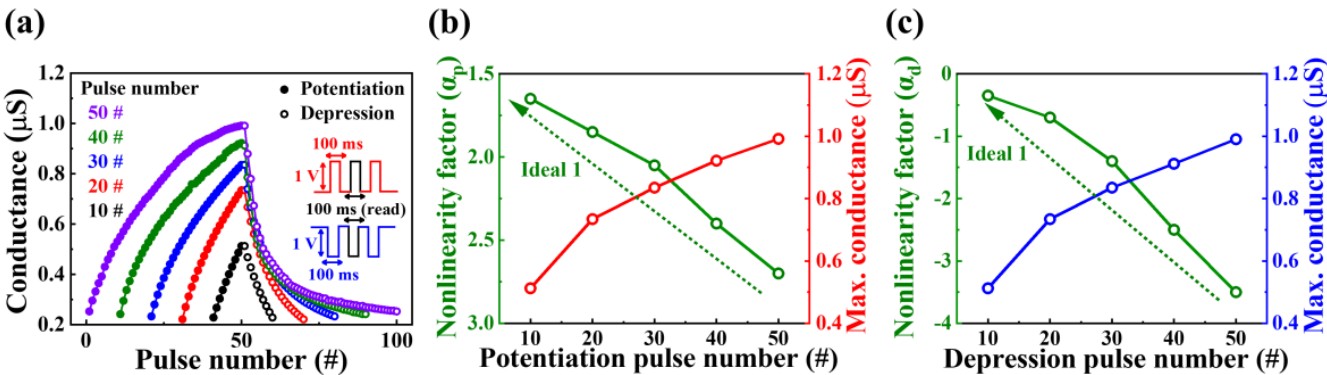

**Figure 7.** (**a**) Potentiation and depression characteristics of synaptic weights by applying pre-synaptic pulses (10# to 50#, steps 10#). Nonlinearity factor and max. conductance of (**b**) potentiation and (**c**) depression as a function of pulse number.

Figure 8a shows the variation in the pre-synaptic pulse number, which was fixed at 30#. Potentiation and depression were induced under three different conditions with varying amplitudes and durations (1 V, 100 ms/1 V, 200 ms/2 V, 100 ms). The results revealed distinct P/D characteristics based on the pre-synaptic conditions. Figure 7b,c respectively show the nonlinearity factors and maximum conductivity values for potentiation and depression with respect to pre-synaptic variables. To summarize the measurement results, it was observed that the nonlinearity factor $\alpha_p$ improved from 2.05 to 1.85 and 1.68, and $\alpha_d$ increased from −1.4 to −0.75 and −0.42 with increasing amplitude and duration. In addition, the maximum conductance for potentiation and depression increased from 0.83 μS to 0.95 μS and 1 μS, and from 0.83 μS to 0.87 μS and 0.96 μS, respectively.

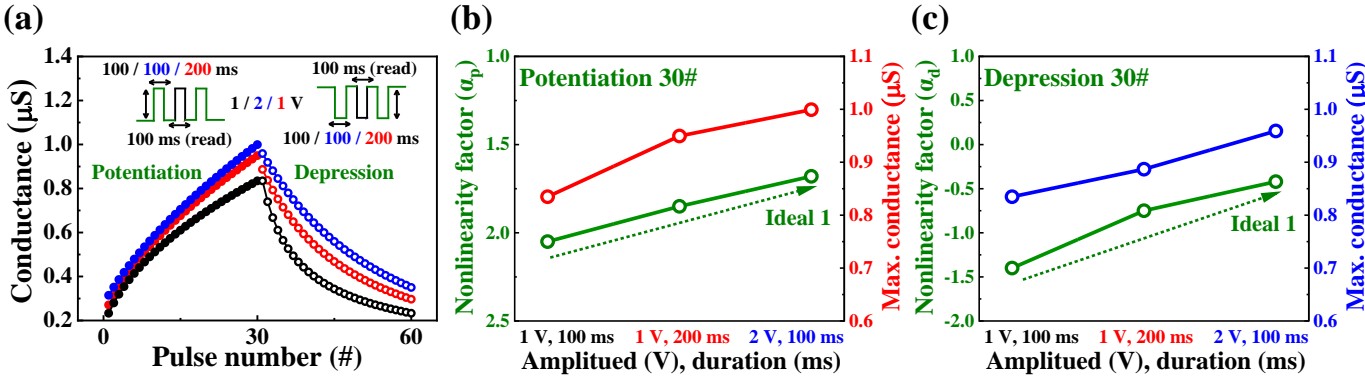

**Figure 8.** (**a**) Potentiation and depression characteristics of synaptic weights following the application of pre-synaptic pulses (1 V, 100 ms/1 V, 200 ms/2 V, 100 ms). Nonlinearity factor and maximum conductance as a function of amplitude and duration for (**b**) potentiation and (**c**) depression.

### 3.5. Recognition Rate of PSG-EDLT in MNIST ANN Simulations

Finally, to explore how the measured P/D characteristics in PSG-EDLT affect the performance of the handwritten digit recognition network built into the software (IBM Analog Hardware Acceleration Kit), we conducted ANN simulations using the MNIST dataset. To normalize the learning and memory characteristics of P/Ds with respect to pulse number, amplitude, and duration, we extracted and analyzed the AR and DR [57]. The AR functions as an indicator of the asymmetry in the conductance characteristics and was extracted using Equation (2):

$$AR = \frac{MAX|G_p(n) - G_d(n)|}{G_p(30) - G_d(30)} \; for \; n = pulse \; spike \; numbers \quad (2)$$

where $G_p(n)$ and $G_d(n)$ represent the conductance of the channel corresponding to the nth pre-synaptic channel. For high learning accuracy, the ideal AR value was 0. DR is an indicator of the relative range of conductance modulation and is defined by Equation (3).

$$DR = G_{max}/G_{min} \quad (3)$$

Theoretically, DR signifies the on/off ratio of the conductance, and a larger value is considered ideal [58]. Figure 9a shows the AR and DR results calculated at an amplitude of 1 V, a duration of 100 ms, and various pulse numbers (10#/20#/30#/40#/50#), as shown in Figure 7a. As the pulse number increased, AR approached the ideal value, decreasing from 0.73 to 0.15, whereas DR increased as a function of the pulse number. Figure 9b shows the AR and DR results for pulse number 30# and different amplitude–duration conditions (1 V, 100 ms/1 V, 200 ms/2 V, 100 ms), as shown in Figure 8a. AR improves from 0.49 to 0.26 at increasing presynaptic strengths, whereas DR exhibits a decreasing trend from 3.58 to 2.86. Figure 9c illustrates the ANN simulations divided into three stages: input, hidden, and output layers, simulating a specific number, 2. As shown in Figure 9d, the

highest recognition rate of 92.19% was observed at pulse number 30# and 1 V/100 ms. Subsequently, under the optimal conditions (pulse number 30#), additional recognition rate measurements were conducted for Figure 9e, which corresponded to 1 V/200 ms and 2 V/100 ms, yielding recognition rates of 92.6% and 93%, respectively. In conclusion, the PSG-EDLT demonstrated recognition characteristics > 90%, thus establishing optimal conditions at 1 V/100 ms, which are crucial for mimicking the synaptic behavior of biological neurons with low-power operations [59].

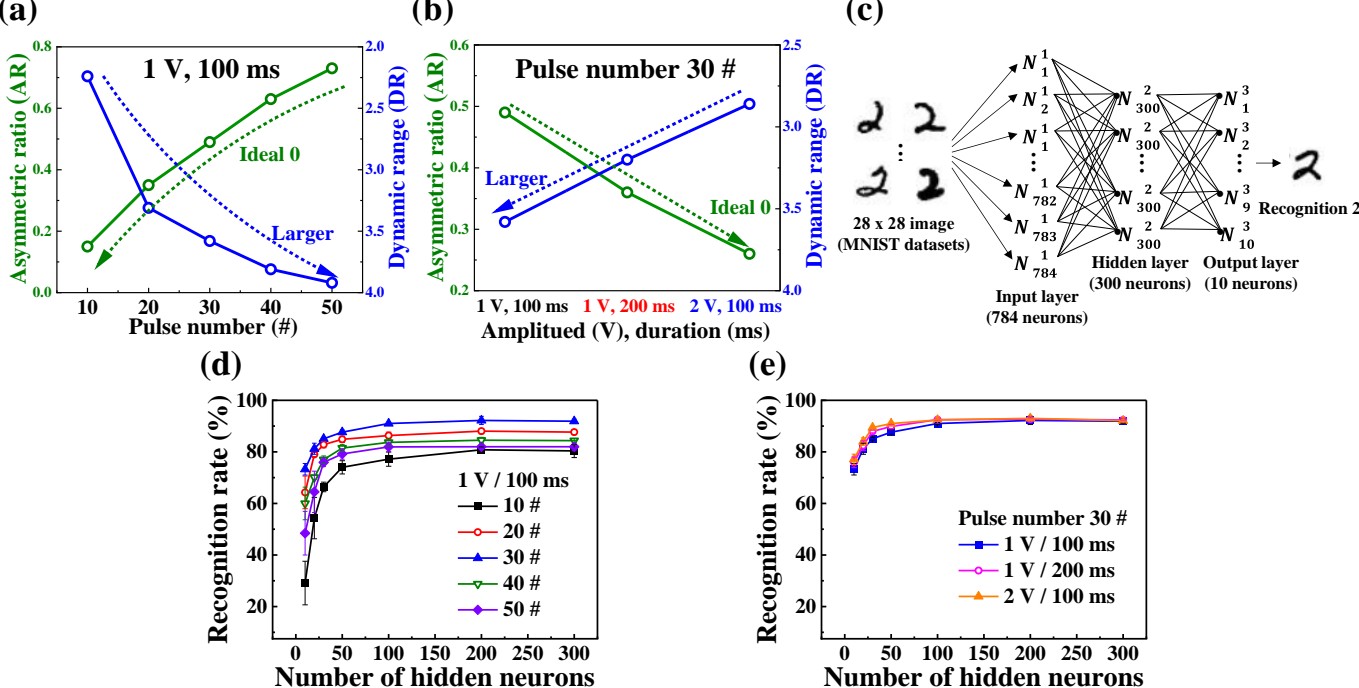

**Figure 9.** Asymmetric ratio and dynamic range of P/D for (**a**) 10# to 50# (fixed 1 V, 100 ms), (**b**) 1 V, 100 ms/1 V, 200 ms/2 V, 100 ms (at 30#). (**c**) Schematic representation of a three-layer fully connected ANN with input, hidden, and output layers for recognition of the modified National Institute of Standards and Technology handwritten digits. Simulated recognition rates for different numbers of hidden neurons of (**d**) 10# to 50# (fixed 1 V, 100 ms) and (**e**) 1 V, 100 ms/1 V, 200 ms/2 V, 100 ms (at 30#). Error bars represent standard deviations.

## 4. Conclusions

This study comprehensively explored the synaptic operational properties of phosphorus-based gate electrolyte EDL materials, distinguishing MOS capacitors and EDLTs that employed PSG and SOG. Analysis of the C–*f* characteristics highlighted PSG's superior capacitance of the PSG over a broad frequency range, thus emphasizing its potential in neuromorphic applications. The precise modulation of the gate voltage elucidated the pronounced counterclockwise hysteresis in the PSG transistors. Comparative analyses involving pre-synaptic, single-spike, double-spike, and frequency-dependent behaviors validated the promising capabilities of the PSG-EDLT and demonstrated its synaptic potential. Extensive evaluations of long-term memory enhancements, learning properties, and ANN simulations further highlighted the outstanding performance of the PSG-EDLT, particularly under optimized conditions. In summary, this study, utilizing PSG films through spin-coating, showcased outstanding synaptic weighting characteristics and demonstrated the potential to achieve a noteworthy recognition rate in handwriting recognition tasks, especially when considering the low-power aspect. These accomplishments not only contribute valuable insights to the field of neuromorphic engineering but also suggest a positive direction for the future development of low-power synaptic devices, thus

demonstrating the potential of phosphorus-based materials in advancing neuro-inspired computing technologies.

**Author Contributions:** Conceptualization, D.-G.M. and W.-J.C.; investigation, D.-G.M. and W.-J.C.; writing—original draft preparation, D.-G.M. and W.-J.C.; MNIST simulation, H.P.; writing—review and editing, D.-G.M., H.P. and W.-J.C.; supervision, W.-J.C.; project administration, W.-J.C.; funding acquisition, W.-J.C. All authors have read and agreed to the published version of the manuscript.

**Funding:** This work was supported by the Korea Institute for Advancement of Technology (KIAT) grant funded by the Korean government (MOTIE) (P0020967, The Competency Development Program for Industry Specialist).

**Data Availability Statement:** The data presented in this study are available in this article.

**Acknowledgments:** The present research has been conducted by the Research Grant of Kwangwoon University in 2023 and the Excellent Research Support Project of Kwangwoon University in 2023. The work reported in this paper was conducted during the sabbatical year of Kwangwoon University in 2023.

**Conflicts of Interest:** The authors declare no conflicts of interest.

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
