# Peer review of "Enhancement of the Synaptic Performance of Phosphorus-Enriched, Electric Double-Layer, Thin-Film Transistors"

_electronics, doi:10.3390/electronics13040737_

Round 1
Reviewer 1 Report
Comments and Suggestions for Authors
In this paper, the author provides a comprehensive comparison between phosphorus-doped silicate glass (PSG) and undoped silicate glass, including the 20B spin-on glass device. The paper effectively demonstrates the advantages and disadvantages of spin-on glass (SOG) devices in both the transfer characteristic curve and synaptic characteristics. Through artificial intelligence simulations using MNIST with PSG devices, the study unequivocally confirms the considerable potential of PSG devices in artificial intelligence applications. However, there are a few issues that need to be addressed before publication.
1. The internal mechanism of artificial synaptic function based on PSG devices requires a more detailed explanation.
2. In Figure 4a of the paper, the almost complete overlap of the two peaks of EPSC makes it challenging to identify the peak of A1. The reported 270.1% increase by the authors needs reconsideration. Providing additional intervals for a more comprehensive comparison is recommended.
3. The reliability of the device is crucial in the application of artificial intelligence. The author should also consider whether the device can exhibit a similar response after multiple potentiation and depression periods.
4. The retention time in Figure 6a is relatively short (approximately 2 s). It is worth exploring whether this can be classified as long-term potentiation (LTP) and depression (LTD).
Comments on the Quality of English LanguageSome sentences need to be polished.
Reviewer 2 Report
Comments and Suggestions for Authors
In this paper, the authors have fabricated synaptic device by P-rich EDLTs and evaluated the performance for neuromorphic systems. The contents seem correct, but I will give the following comments.
(1) First, what is the improvement realized by your device ? I think that accuracy of 93% in MNIST is not so high in comparison with other method.
(2) What is the physical mechanism of the hysteresis in Fig. 2(a) ? It is preferable to show it by some figure.
(3) As for the ANN, first, please tell the detailed circuit implementing the EDLTs. Moreover, tell how the synaptic weights is determined, how they are converted to the EDLT characteristic, and how the characteristic is controlled in practical way. Furthermore, "2 300" "2 300" ... should be "2 1" "2 2" ... ?
Reviewer 3 Report
Comments and Suggestions for Authors
Authors studied Phosphorus-doped silicate gasses (PSG) novel solid electrolyte as gating medium for ion-gated transistors. They demonstrate that devices based on IGZO semiconductor channel achieve high modulation efficiency, while transient experiments demonstrate potential applications towards neuromorphic electronics. In my assessment, the work is competently reported and can help advancing systems of technological interest. However, a few points should be addressed before publication:
1- Although the introduction is very complete on applications of neuromorphic systems and how the experiment and interpretation was devised, a section about latest research on novel solid electrolytes for neuromorphic application is still missing. The authors should include a section about that to highlight what is the impact of the findings in the article regarding the new PSG electrolyte they are proposing.
2- Some important curves for new transistor benchmarking are missing. The authors should include Transistor Output characteristics (ID vs Vd at constant Vgs steps) in linear and saturation regime.
3-Likewise, authors should include discussion of the charge carrier density and mobility achieved using the different electrolytes.
4- Authors should compare the figures-of-merit achieved for their new electrolyte with other works that gated IGZO and discuss the advantages.
Round 2
Reviewer 1 Report
Comments and Suggestions for Authors
Good.
Author Response
Thank you for positively reviewing the revised manuscript.
Reviewer 3 Report
Comments and Suggestions for Authors
After reading the response to the authors and the revised document, I am concerned that some of the issues raised were not revised according to the suggestions. Therefore, the issues remain before the publication can be recommended.
- In response to comment 2: the authors write that “In response to the reviewer's insightful suggestion, we have revised the manuscript to incorporate Transistor Output Characteristics”. However, they do not point to the new figures of ID vs Vd output curves required for that. As I pointed out in my previous revision, this type of figure is fundamental for the reporting of ion-gated transistors and for the discussion of the linear and saturation behavior.
- In response to comment 3: The authors did not discuss the charge carrier density and mobility, as pointed out in comment 3. They mentioned that “(…) have revised the manuscript to include a more comprehensive discussion on charge carrier density and mobility, (…)”, but they do not mention which paragraphs were revised and didn’t include the calculations of these two benchmark figures-of-merit.
In response to comments 1 and 4: the manuscript is still lacking a proper revision of what are the newer solid electrolytes proposed in literature for electrolyte gating and proper positioning how the new electrolyte PSG proposed in this paper is a relevant innovation regarding figures-of-merit. I am afraid that the manuscript cannot be accepted if the level of advancement to the field is not well accessed
Round 3
Reviewer 3 Report
Comments and Suggestions for Authors
After the previous round of revisions, the manuscript can be accepted for publication